# Kinesiophobia in Injured Athletes: A Systematic Review

**DOI:** 10.3390/jfmk9020078

**Published:** 2024-04-19

**Authors:** Jatin P. Ambegaonkar, Matthew Jordan, Kelley R. Wiese, Shane V. Caswell

**Affiliations:** Sports Medicine Assessment Research & Testing (SMART) Laboratory, George Mason University, Manassas, VA 20110, USA; mjorda@gmu.edu (M.J.); kwiese2@gmu.edu (K.R.W.); scaswell@gmu.edu (S.V.C.)

**Keywords:** Tampa Scale of Kinesiophobia, fear of reinjury, fear of movement

## Abstract

Athletes have a high risk of injury. Kinesiophobia is a condition in which an individual experiences a fear of physical movement and activity after an injury occurs. Our purpose was to systematically review the literature about Kinesiophobia in athletes. A systematic review was conducted in February 2023 using PubMed, CINAHL, SPORTDiscus, Web of Science, Cochrane Library, and Medline. Studies were included if they were peer-reviewed, in English, within the last 20 years and included athletes who had been injured and tracked Kinesiophobia. Articles were checked for quality via the modified Downs and Black checklist. Fourteen studies were included in the review and had an average “fair” quality score. Authors examined Kinesiophobia in injured athletes with mostly lower-extremity injuries. Kinesiophobia was associated with lower physical and mental outcomes. Kinesiophobia exists in athletes and can affect both physical and mental factors. The Tampa Scale of Kinesiophobia (TSK) was the most common tool used to examine Kinesiophobia. Common mental factors associated with Kinesiophobia include anxiety, low confidence, and fear avoidance.

## 1. Introduction

Approximately 8.6 million sports-related injuries occur every year [1]. Sports-related injury can result not only in physical disability, but may also have psychological impacts [2,3]. Kinesiophobia is a psychological concept that affects the athletic population and can have a negative impact on rehabilitation progression and return to sport [3]. Kinesiophobia is defined as an irrational and debilitating fear of physical movement and activity resulting from feeling vulnerable to painful injury or reinjury [4]. This fear consequently affects the athlete both physically (e.g., decreased muscular strength, impaired proprioception, and decreased range of motion) [5,6,7,8] and psychologically (e.g., anxiety, depression, and decreased health-related quality of life) [7,8,9,10]. Fear of movement tends to increase pain-related fear and can be associated with safety-seeking behaviors, such as the avoidance of certain movements [7].

Authors have previously used the terms Kinesiophobia, fear of movement, and fear of reinjury interchangeably in previous literature [3,4,11]. For the purpose of this article, fear of movement and fear of injury are separately, operationally defined in regard to Kinesiophobia. Previous authors have described a fear of movement as occurring at the early stage post-injury in which the patient is hesitant to perform a basic movement, such as walking [3]. Fear of reinjury is commonly used during the later stages of rehabilitation where the patient is hesitant to participate in functional athletic movements (e.g., cutting) [3]. Fear of reinjury can be triggered in settings in which the athlete was initially injured [3].

Athletes who are experiencing Kinesiophobia are likely to experience reduced physical function, affecting their ability to progress through rehabilitation programs and their quality of life [9]. In some cases, Kinesiophobia is reported to negatively affect functional outcomes because patients may be hesitant to complete triggering rehabilitation exercises, delaying the recovery process and leading to decreased strength and range of motion [6]. However, Kinesiophobia may be overlooked because practitioners may not be aware of the concept or they may assume the athlete is eager to return to play [4].

The fear avoidance model [7] explains how and why injuries can result in Kinesiophobia and other factors, such as chronic pain. When an athlete suffers an injury and experiences pain, they either have high or low catastrophization, which determines their fear levels [7]. Low fear levels allow the athlete to interpret the pain as non-threatening, promoting normal recovery [7]. However, if an athlete perceives the pain as threatening, likely causing a fear of movement, it can lead to Kinesiophobia [7]. Kinesiophobia can present in many individuals either post-injury or following surgery, but the length of time in which Kinesiophobia persists varies across individuals [11,12,13,14,15,16]. Irrespective of its onset, Kinesiophobia complicates a full return to participation in sport [11,15,16]. Prior authors note that less than 50% of athletes return to pre-injury activity levels [14,15,17]. Furthermore, fear of movement and/or fear of reinjury can delay the Return-to-Play (RTP) process and may negatively impact rehabilitation outcomes. For example, a fear of movement may lead to decreased muscular strength, increased postural sway, and impaired proprioception, perpetuating chronic conditions that hinder an athlete’s athletic ability [5,8].

Overall, despite the existence of Kinesiophobia and the negative outcomes associated with Kinesiophobia, relatively limited literature exists describing the presence of Kinesiophobia in athletes and current practices to address Kinesiophobia. This gap in the literature is problematic because clinicians may not know how to properly rehabilitate and return athletes who have a fear of movement or reinjury. Additionally, an awareness of Kinesiophobia allows the healthcare team to implement objective Kinesiophobia measures into rehabilitation protocols and ensure the athlete possesses the confidence and psychological readiness to return to play. Thus, the purpose of this study was to systematically review the current literature examining Kinesiophobia in injured athletes.

## 2. Materials and Methods

### 2.1. Search Strategy and Study Selection

This review was conducted in accordance with the Preferred Reporting Items for Systematic Reviews and Meta-Analyses (PRISMA) guidelines [18]. Six electronic databases were systematically searched through 25 February 2023, including PubMed, CINAHL, SPORTDiscus, Web of Science, Cochrane Library, and Medline. Articles were included if they were published within the last 20 years to ensure the evidence was current and relevant. The inclusion criteria and exclusion criteria that were applied to this review can be seen in Table 1, and the search strategy and terms used can be found in Table 2.

### 2.2. Data Extraction

A two-part screening process was implemented following the initial search. First, two investigators screened article titles and abstracts to determine whether they were relevant to the scope of the review. Following, the full text of the articles was examined to determine inclusion and exclusion eligibility. A third expert reviewer resolved any disagreement or discrepancy to determine article inclusion and exclusion.

### 2.3. Methodological Rigor and Study Quality Assessment

The modified Downs and Black (mDB) checklist appraisal tool was used to assess the methodological rigor and study quality for the chosen articles [19]. This appraisal tool was designed to assess both randomized and non-randomized studies [19]. The mDB checklist consisted of 27 questions, separated into 5 categories (reporting, external validity, internal validity—bias, internal validity—confounding, and power), including how to score each question [19].

## 3. Results

### 3.1. Study Selection

During the initial literature search, 41 studies were screened. A total of 14 studies fit the inclusion criteria and are included in this review. The overall purpose of the included articles was to examine the presence of Kinesiophobia in injured athletes or use Kinesiophobia as a patient-reported outcome measure to examine the change over time (see Figure 1 depicting the PRISMA flowchart).

### 3.2. Methodological Rigor and Study Quality Assessment

About half of the studies in this review were of higher quality (>71.4%). The highest scores on the MDB checklist were 27/28 and 20/28 [11,16,20,21,22,23] (see Table 3). All included studies directly stated the objective/aim, characteristics of participants, outcome measures, and main findings. Only two of the included studies described the intervention of interest [20,21]. Most of the studies did not have a treatment or placebo and were rather simply observing measures over time. Only one study [20] reported possible adverse events. Most studies reported participants lost to follow-up. External validity was determined to be overall good quality, with 11/14 studies scoring 3/3 within the category. Scores for internal validity—bias were mixed, due to subjects and researchers not being blinded in most studies. Internal validity—confounding results were mixed as well, due to the questions about randomization not being applicable to most of the included study designs. All but two studies [24,25] scored 1/1 for the power category. The lowest scores on the mDB checklist were 13/28 and 14/28 [24,25].

### 3.3. Participant Characteristics

Participant characteristics are presented in Table 4. The researchers examined Kinesiophobia in both males (*n* = 561) and females (*n* = 423). The level of sport participation varied in the 14 studies included in the review. In one study, the authors examined adolescent athletes [20], three examined high-school and/or collegiate athletes [21,22,23], one examined professional athletes [24], two examined recreational athletes [5,25], and seven examined a combination of levels [11,16,26,27,28,29,30], such as recreational and collegiate athletes. The athletes played diverse sports, including running [25], football and lacrosse [21], alpine skiing [24], and various college sports, including baseball, basketball, futsal, gymnastics, lacrosse, soccer, softball, table tennis, tennis, and track and field [23]. Most studies examined athletes who had anterior cruciate ligament or other knee injuries [16,24,25,26,27,28,29], or ankle injuries [5,21,23,30].

### 3.4. Objective Measures of Kinesiophobia

The authors used several tests (see Table 5) and objective physical measures to assess Kinesiophobia (see Table 6) including joint-position sense [5], postural control [5], strength [11,29], joint laxity [21,29], muscle activity [29], and performance-based functions [11,16,20,26,29,30]. We found that authors commonly use performance-based functions, often via horizontal hops tests for distance (single leg or double leg), side to side hops, heel raises, and/or by examining peak vertical ground reaction forces. For example, Alshahrani et al. [5] examined how Kinesiophobia might affect ankle joint-position sense and found a significant positive correlation with ankle joint-position sense errors both in dorsiflexion and plantarflexion, as well as with postural control. Ohji et al. examined peak vertical ground reaction force and found no significant correlations between the vertical ground reaction force and TSK-11 scores. However, they found that vastus medialis muscle activity, while landing from a jump, was positively correlated with TSK-11 scores [29]. Finally, Paterno et al. found that patients who had higher TSK-11 scores were more likely to have a quadricep muscle strength symmetry and a hop limb symmetry lower than 90% [11]. Kinesiophobia had a high correlation with a fear of reinjury [11,21,22,28,29,30], fear of movement [5,21,23,26], and confidence levels [16,23,24,26,30] in lower limb movement. Other objective outcome measures previously used to assess Kinesiophobia include activity level [11], injury tracking [11], and reliability and validity of the TSK [27].

### 3.5. Subjective Measures of Kinesiophobia

Kinesiophobia can be measured subjectively using several surveys (see Table 5), including the Athlete Fear Avoidance Questionnaire (AFAQ) [21], the Reinjury Anxiety Inventory (RIAI) [22], the Tampa Scale of Kinesiophobia (TSK) [5,20,21,22,24,25,26,27,28,30], the TSK-11 [11,22,23,29], and the TSK-17 [16]. The TSK-17 is the standard scale, consisting of a 17-item checklist that has statements regarding fear of movement, reinjury, and fear-avoidance in which participants use a 4-point Likert scale to rate how much they agree or disagree with each statement [31]. The TSK-11 is a shortened version the TSK-17, consisting of 11 items rather than 17, and is used more commonly [31].

**Table 5 jfmk-09-00078-t005:** Tests used to assess Kinesiophobia in injured athletes.

Study	Year	Test
Alshahrani [5]	2022	TSK
Bagheri [25]	2021	TSK
Fukano [21]	2020	AFAQ
Hart [26]	2019	TSK
Houston [22]	2014	TSK-11
Huang [27]	2019	TSK
Jedvaj [24]	2021	TSK
Kvist [28]	2004	TSK
Ohji [29]	2022	TSK-11
Paterno [11]	2018	TSK-11
Reinking [20]	2022	TSK
Slagers [30]	2021	TSK
Theunissen [16]	2019	TSK-17
Watanabe [23]	2023	TSK-11

**Table 6 jfmk-09-00078-t006:** Physical measures analyzed examining studies about Kinesiophobia in injured athletes.

Outcome Measure	Study	Specific Measure; Units
Ankle joint-position sense	Alashahrani 22 [5]	Dual digital inclinometer, degrees
Postural control	Alashahrani 22 [5]	Stabilometric force platform, mm squared
Knee symptoms and function	Bagheri 21 [25]	KOOS-ADLs and KOOS sports activities scale, 0–100
Joint laxity	Fukano 20 [21]	Ankle arthrometer, degrees
	Ohji 22 [29]	KT-1000, degrees
Functional instability	Fukano 20 [21]	Identification of functional ankle instability score, score
Performance-based function	Hart 19 [26]	Hops for distance, cm; side to side hops in 30 s, number; cross-over hop for distance, cm
	Ohji 22 [29]	SL hop distance, cm; SL jump landing: peak vertical ground reaction force, N; time to peak force; s
	Paterno 18 [11]	SL hop for distance, cm; triple hop for distance, cm; triple cross-over hop for distance, cm; 6 m timed hop, cm; limb symmetry index, %
	Slagers 21 [30]	SL heel-raise test for endurance; number; SL hop test for distance; cm, limb symmetry index, %
	Reinking 22 [20]	Reaction time, ms
	Theuniessen 19 [16]	IKDC-2000 score, 0–100
Strength	Ohji 22 [29]	Biodex system 4 (peak torque)-measured isokinetic knee strength, N
	Paterno 18 [11]	Biodex isokinetic dynamometer-measured isometric quadricep femoris strength (peak torque), N
Muscle activity	Ohji 22 [29]	sEMG, Root Mean Square Activation (%maximum voluntary isometric contraction)

KOOS = Knee Injury and Osteoarthritis Outcome Score, ADL = Activities of Daily Living, SL = Single Leg; sEMG = Surface Electromyography.

Nine of the 14 articles only assessed a single measurement of Kinesiophobia using a survey [5,11,21,23,24,26,27,28,29]. The other five articles implemented a repeated measures design in which participants completed a survey multiple times (two to three) to examine the change in subjective Kinesiophobia levels over time [16,20,22,25,30]. Pain was also examined in several studies as an outcome measure, usually via a visual analog scale or patient-reported outcome measure questionnaires [16,23,25,26,29,30].

Ten studies examined how Kinesiophobia affected athletes psychologically, specifically at the time of RTP and beyond [11,16,20,21,22,23,24,26,28,30]. Researchers assessed many different psychological outcome measures, including Kinesiophobia [5,16,20,23,29,30], fear of movement/reinjury [21,22,24,25,26,28], patient reported fear [11,21,22], coping strategies [25], confidence [26], and anxiety [22] (see Table 7). Houston et. al, Reinking et al., Slagers et al., and Theunissen et al. examined how psychological symptoms of Kinesiophobia changed over time [16,20,22,30]. Overall, these four studies found that, as the athlete’s physical symptoms improved over time during rehabilitation, Kinesiophobia and a fear of reinjury decreased for the majority of participants [16,20,22,30]. Individuals with mild to moderate musculoskeletal injuries experienced a significant improvement in TSK-11 and RIAI scores 3 weeks post-injury [22]. However, in individuals with an Achilles tendon rupture that were still psychologically impacted by Kinesiophobia 6 months post-injury, the presence of symptoms determined the amount of physical activity they were willing to complete [30]. In contrast, post-operative ACL reconstruction (ACLR) surgery patients were found to have a decreased level of Kinesiophobia 12 months following surgery, with the number of ACLR patients reporting high levels of Kinesiophobia decreasing by about 61% (92 to 36 patients) [16].

Huang et al. examined the validity and reliability of the TSK, specifically the Japanese TSK (TSK-J), in patients with ACLR, and found good reliability but low validity and responsiveness [27]. They suggested that the TSK-J may not the best way to assess psychological factors in patients with ACL injuries [27]. Other patient-reported outcome measures include the visual analog scale (VAS) [32] for pain and the disablement in the physically active scale (DPAS) [8]. The VAS is used to track patients’ pain progression or compare pain severity between patients with similar conditions [32]. The VAS can be administered using numerical rating scales, graphic rating scales, or curvilinear scales, and patients mark the point on the line that they feel represents their perception of pain [32]. While the VAS does not explicitly measure Kinesiophobia or fear, it may be a good tool to use in combination with the TSK for clinicians to track pain alongside fear levels. The DPAS is a tool that measures the level of disablement in physically active populations [8]. It consists of 16 items that assesses both physical health and mental health [8]. Higher scores indicate greater levels of disablement [8].

### 3.6. Other Measures of Kinesiophobia

Researchers in six studies assessed a one-time measurement of Kinesiophobia and used those scores, along with other outcome measures, to assess for correlations between outcome variables [11,21,23,24,26,28]. Fukano et al. compared TSK and AFAQ scores in individuals with functional ankle instability (FAI) to individuals who had sprained their ankle previously, but were not diagnosed with functional ankle instability (NFI) [21]. Individuals with FAI had higher TSK scores compared to those without functional ankle instability [21]. As a result, the authors concluded that the presence of an FAI could be associated with a higher level of fear of movement and reinjury [21].

Similarly, Watanabe et al. concluded that even a perceived instability with FAI patients may be related to Kinesiophobia [23]. Kvist et al. reported a weak negative correlation between the TSK and present pain, but patients who did not return to their pre-injury activity levels following ACLR had more fear of pain or reinjury [28]. This trend of patients not returning to their pre-injury activity levels was also observed by Paterno et al. and Hart et al. [11,26]. Psychological readiness to RTP and knee confidence are two factors that can determine whether an ACLR athlete is psychologically ready to return to sport or even perform specific movements, and could contribute to an athlete’s ability to return to their to pre-injury activity levels [26].

Bagheri et al. conducted a randomized controlled trial on female recreational runners with patellofemoral pain syndrome (PFPS) to compare treatments of only exercise versus a combination of exercise and mindfulness [25]. The group that completed the mindfulness training, consisting of breathing, meditation, yoga, and stress reduction, reported a decreased fear of movement following the intervention [25].

## 4. Discussion

### 4.1. Primary Findings

The primary findings of this systematic review reveal that Kinesiophobia exists in athletes both physically and psychologically. The TSK is the most common tool in the literature to assess subjective accounts of Kinesiophobia. Psychological factors associated with Kinesiophobia include anxiety, confidence, and fear avoidance.

### 4.2. Methodological Rigor and Study Quality Assessment

The average score of the studies was 65%, or 18 points, which is a “fair” score [19]. Reporting items within the studies were described in most of the studies, and external validity was present in all but three studies. Still, given the relatively low sample sizes of studies in this review, we believe that additional longitudinal examinations are needed to examine the associations of Kinesiophobia with return-from-injury timelines in injured athletes. The articles in this review include cross-sectional, prospective cohort, and a randomized controlled trial. This finding indicates that there is an increasing interest in the area with researchers examining Kinesiophobia in injured athletes using multiple types of study designs.

### 4.3. Characteristics of Included Studies and Participant Demographics

The range of ages of athletes included in the studies was 15~42 years old. Across the studies, both male and female athletes were examined across many different levels of sport. Only one of the studies suggested that females had a higher chance of reporting higher TSK-11 scores [23], but there were only five females included in that particular study compared to 37 males. This ratio of females to males in this study made it difficult to make conclusive statements on the differences in Kinesiophobia levels between sexes. The majority of the researchers examined Kinesiophobia in athletes with lower-extremity injuries. Specifically, several authors examined Kinesiophobia in athletes with knee injuries, with anterior cruciate ligament (ACL) injury being the most common knee injury, supporting the idea that ACL injury and reconstuction are extensively associated with Kinesiophobia [16]. Several authors also examined Kinesiophobia in athletes with ankle instability, which is understandable given that a lateral ankle sprain is the most prevalent lower-extremity musculoskeletal injury in physically active individuals [33].

### 4.4. Tests Used to Assess Kinesiophobia

The TSK survey was most consistently used to measure Kinesiophobia. Although Huang et al. [27] indicate that the Japanese version of the TSK (TSK-J) may not the best way to assess psychological factors for patients with ACL injuries, most other researchers indicate the TSK as a means to objectively measure Kinesiophobia. We found that the TSK is the most popular measurement tool to assess Kinesiophobia because it is based on the fear avoidance model and has been found to be valid and reliable [27,34]. The TSK-11 is suggested for use with athletes because of its high reliability and satisfactory validity [31], but it is also a condensed version of the TSK. Thus, it does not take as much time for completion, increasing compliance. The shortened TSK-11 is also beneficial when athletes are completing it multiple times.

Other surveys, like the AFAQ, measure injury-related fear avoidance and can be taken alongside the TSK to provide a comprehensive understanding of any mental barriers an athlete is facing pertaining to fear of movement or reinjury [21]. Similar to the VAS, the DPAS may be a useful tool to incorporate alongside the TSK as the scale does not measure fear levels directly. By using these three surveys in conjunction with one another, clinicians can understand how the athlete perceives their fear, ability, and pain.

### 4.5. Physical Measures to Assess Kinesiophobia

Kinesiophobia was found to have negative impacts on strength and postural control [5,11,29]. Based on this information, there is a chance that an athlete who has high levels of Kinesiophobia will have resulting functional deficits. This idea can be tied to the fear avoidance model, where a high catastrophization of pain leading to high anxiety of pain perpetuates a cycle of a fear of movement [7]. This fear causes an avoidance of movement, which can inhibit the muscles, tendons, and ligaments around the area, thus leading to muscle atrophy, fibrosis, and functional impairment [5]. As a result, altered motor patterns occur, and can lead to decreased strength and postural control in the affected area [5].

Kinesiophobia is also associated with diminished performance-based function [11,16,20,26,29]. Performance-based function, or how well an athlete can perform an advanced set of movements, is related to the functional demands of their sport. Performance-based function aligns with Kinesiophobia more commonly as an athlete is closer to returning to a sport [3]. High Kinesiophobia and fear of reinjury levels can cause an athlete to reduce their exposure to physical activities, especially those in which they can possibly reinjure themselves, leading to a perception of limited function or an actual decrease in performance-based function [3]. This finding supports the importance for clinicians to track Kinesiophobia in their athletes to help address it, so that performance and functional levels do not continue to decrease. If Kinesiophobia is left unaddressed, everyday functional activities could be affected [9].

### 4.6. Limitations and Future Recommendations

We acknowledge some study limitations. First, despite using a comprehensive search strategy, we recognize that some relevant studies may have been excluded. For example, we did not find studies assessing Kinesiophobia for athletes with upper-extremity injuries, with only one study examining musculoskeletal injuries irrespective of location [22]. Additionally, there was an inconsistency in athlete level in the reviewed articles. Future researchers should assess athletes across levels (e.g., high school, collegiate, and professional) to understand how Kinesiophobia affects athletes at various levels when returning to play.

We also note the need for additional research to examine how Kinesiophobia affects athletes across several sports, since a majority of the included studies (11) did not report which sport was assessed. The information is needed because Kinesiophobia levels may vary across sports and athletic activities that involve contact with other players (e.g., soccer and wrestling) versus non-contact sports (e.g., tennis, and track and field). Therefore, the results of this review cannot be directly generalized to all types of athletes across levels and types of sport.

Future researchers should also examine treatment options for Kinesiophobia to identify the options that are most effective for addressing Kinesiophobia in athletes. It is important to note that none of the included articles described how effective repeated use over time was when using the TSK. Furthermore, only one study stated the minimal clinically important difference with the TSK, which was reported as a score of 4 [16]. However, this was only in regards to patients with low back pain [16]. Therefore, future researchers should examine minimal clinically important difference values with the TSK as well. This work can allow clinicians the opportunity to document meaningful objective measurements during the return-to-play process.

### 4.7. Clinical Implications and Applications

The primary clinical implication of the current study is that clinicians should be aware of the potential presence of Kinesiophobia in athletes post-injury. It is important for practitioners to monitor Kinesiophobia scores throughout the rehabilitation process to monitor both psychological and physical recovery in athletes to prevent a decrease in quality of life during the return-to-play process.

Furthermore, it is important to educate athletes, coaches, and the multidisciplinary healthcare team caring for the athletes about Kinesiophobia. This education could reduce the athletes’ anxiety [3], and if all stakeholders (athletes, parents, coaches, and healthcare practitioners) are educated about Kinesiophobia and the anticipated symptoms, then everyone supporting the athlete through recovery may be able to recognize and address early signs of Kinesiophobia that could hinder the injury recovery process. If coaches know how to recognize Kinesiophobia-related signs that are diminishing an athlete’s performance, they can communicate that to the athletic trainers and healthcare team. The healthcare team can then work with the athlete to overcome his/her fear. Likewise, if athletes are able to recognize and articulate their symptoms of Kinesiophobia, they can communicate their mental and physical barriers that may be inhibiting their optimal performance. Overall, once practitioners are equipped to recognize the signs of Kinesiophobia, they can integrate appropriate techniques into treatment strategies to proactively assess and address Kinesiophobia.

Practitioners can use the TSK as a means to objectively measure Kinesiophobia. The TSK is currently the only tool that specifically aims to measure Kinesiophobia [31]. The current review findings indicate that the TSK-11 is the preferred form of the TSK to use because it has high reliability and high validity compared to other versions [31]. The shortened TSK-11 also allows multiple administrations to objectively measure psychological Kinesiophobia feelings throughout the rehabilitation process.

In addition, the whole sports medicine team (e.g., athletic trainers, physical therapists, physicians, coaches, and others) can create a plan to address Kinesiophobia. This plan can include mindfulness or relaxation techniques that could reduce tension and anxiety [3,25]. The team can also work with the athlete to set goals, which provides the athlete with direction and the ability to visualize the progress that is made during rehabilitation [3]. Graded exposure may also be an effective technique to gradually expose the athlete to fearful movements to decrease Kinesiophobia levels [3]. Furthermore, appropriate social support may enhance the athlete’s coping strategies [3]. Implementing education, recognition, assessment, and appropriate plans for athletes with Kinesiophobia will support athletes in overcoming their fears.

Overall, Kinesiophobia levels should be considered as an essential return-to-play criteria similar to pain, range of motion, and strength measurements. The current review provides evidence that there is an increasing amount of interest in the topic of Kinesiophobia in injured athletes, evidenced by the finding that, in the final included articles, almost all (13 of 14) of them were conducted within the last 10 years. Clinicians should implement proper education, recognition, assessment, and plan to help athletes with Kinesiophobia to overcome the condition. This education about Kinesiophobia can help clinicians, coaches, and athletes become aware of the condition so they know how to identify who may have Kinesiophobia, ultimately helping athletes become less fearful and gain confidence when recovering from an injury. 

## 5. Conclusions

The current findings indicate that Kinesiophobia exists in athletes and can affect both physical and mental factors. The Tampa Scale of Kinesiophobia is the most common survey tool used to measure Kinesiophobia. Common psychological factors associated with Kinesiophobia include anxiety, confidence, and fear avoidance.

## Figures and Tables

**Figure 1 jfmk-09-00078-f001:**
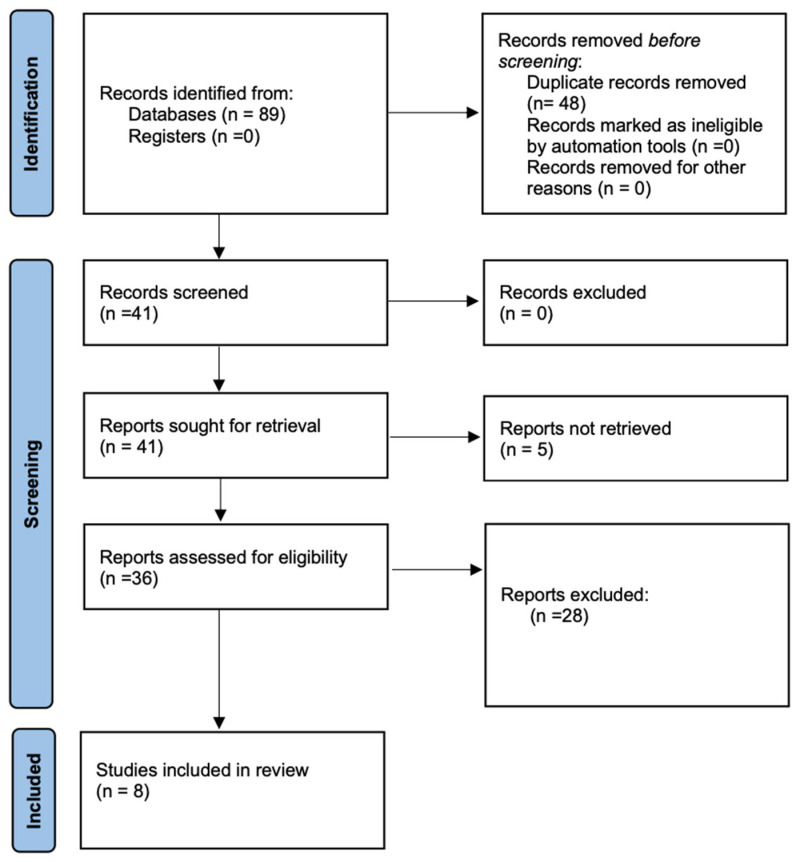
PRISMA flowchart of studies about Kinesiophobia in injured athletes.

**Table 1 jfmk-09-00078-t001:** Inclusion and exclusion criteria in studies examining Kinesiophobia in injured athletes.

Inclusion Criteria	Exclusion Criteria
Athletes who have been injured	Reviews
Track Kinesiophobia	Case studies
Peer-reviewed	Conference proceedings
English articles	
Published within last 20 years	

**Table 2 jfmk-09-00078-t002:** Search strategy and search terms used to examine Kinesiophobia in injured athletes.

Step	Search Terms	Boolean Operator	PubMed	CINAHL Plus	Sport Discus	Web of Science	Cochrane Library	MedLine
1	Kinesiopho *		140	925	439	1459	874	1446
2	Injur *		108,825	368,476	169,496	952,027	71,764	1,417,231
3	athlet *		9353	80,516	397,997	95,466	11,813	111,143
4	Reinjur *		172	1477	579	943	259	1207
5	Fear *		10,450	51,694	10,568	156,481	11,475	122,033
6	Moveme *		54,609	96,129	85,431	732,726	39,633	637,703
7	1, 2	AND	7	44	42	47	10	66
8	7, 4	AND	2	12	9	10	4	13
9	5, 6	AND	341	1474	685	5980	915	3695
10	7, 9	AND	2	18	17	17	2	0
11	1, 2, 4,	AND, OR	32	216	0	379	144	338
12	11, 5, 6	AND, OR	23	152	0	312	89	235
13	11, 9	AND	14	99	0	229	49	146
14	1, 5, 6,	AND, OR	439	2144	973	6885	1590	4697
15	1, 4, 5	AND, OR	157	1062	495	1555	907	1555
16	14, 3	AND	13	981	504	1522	887	1508
17	15, 3	AND	16	1002	471	1506	880	1504
18	1, 3, 2, 4, 9	AND, OR	13	18	17	19	2	20

The asterisk sign indicates a truncation of the word and allows a wildcard search for all the variable endings of the root word.

**Table 3 jfmk-09-00078-t003:** Methodological rigor of studies examining Kinesiophobia in injured athletes using the modified Downs and Black (mDB) criteria.

Study	Reporting	External Validity	Internal Validity—Bias	Internal Validity—Confounding	Power	Total	%
Alshahrani 22 [5]	6	3	5	1	1	16	57.1
Bagheri 21 [25]	10	3	7	6	1	27	96.4
Fukano 20 [21]	7	1	5	1	0	14	50.0
Hart 19 [26]	7	0	5	2	1	15	53.6
Houston 14 [22]	8	3	5	3	1	20	71.4
Huang 19 [27]	7	3	3	4	1	18	64.3
Jedvaj 21 [24]	7	3	4	3	1	18	64.3
Kvist 04 [28]	6	3	5	3	1	18	64.3
Ohji 22 [29]	6	1	4	2	0	13	46.4
Paterno 18 [11]	7	3	5	4	1	20	71.4
Reinking 22 [20]	7	3	5	4	1	20	71.4
Slagers 21 [30]	7	3	5	4	1	20	71.4
Theunissen 19 [16]	7	3	5	4	1	20	71.4
Watanabe 23 [23]	6	3	4	2	1	16	57.1

**Table 4 jfmk-09-00078-t004:** Participant characteristics in studies examining Kinesiophobia in injured athletes.

Study	Year	Training Level	Injury	Mean Age (y)	Sport	Female (*n*)	Male (*n*)	Total (*n*)
Alshahrani [5]	2022	Recreational	Functional Ankle Instability	23	Not Reported	21	34	55
Bagheri [25]	2021	Recreational	Patellofemoral Pain	28.35	Running	33	0	33
Fukano [21]	2020	Collegiate	Functional Ankle Instability	19.45	Football and Lacrosse	105	79	89
Hart [26]	2019	Athletes (Various Levels)	Anterior Cruciate Ligament	31	Not Reported	42	76	118
Houston [22]	2014	High School and Collegiate	Acute Musculoskeletal Injury (Inability to Fully Participate in Sport for at Least 2 Days)	17.9	Not Reported	11	11	22
Huang [27]	2019	Athletes (Various Levels)	Anterior Cruciate Ligament	32.4	Not Reported	81	141	222
Jedvaj [24]	2021	Professional	Knee Injury	24	Alpine skiing	22	11	33
Kvist [28]	2004	Athletes (Various levels)	Anterior Cruciate Ligament	27	Not Reported	28	34	62
Ohji [29]	2022	Athletes (Various Levels)	Anterior Cruciate Ligament	20	Not Reported	13	18	31
Paterno [11]	2018	Athletes (Various Levels)	Anterior Cruciate Ligament	16.2	Not Reported	Not Reported	Not Reported	40
Reinking [20]	2022	Adolescent	Concussion	15.85	Not Reported	24	25	49
Slagers [30]	2021	Athletes (Various Levels)	Achilles Tendon Rupture	42.6	Not Reported	16	34	50
Theunissen [16]	2013	Athletes (Various Levels)	Anterior Cruciate Ligament	30.5	Not Reported	43	59	102
Watanabe [23]	2023	Collegiate	Chronic Ankle Instability	20.5	Badminton, Baseball, Basketball, Futsal, Gymnastics, Lacrosse, Soccer, Softball, Table Tennis, Tennis, and Track and Field	5	37	42

**Table 7 jfmk-09-00078-t007:** Psychological measures analyzed in studies examining Kinesiophobia in injured athletes.

Outcome Measures	Study	Specific Measure
Kinesiophobia/fear of movement or reinjury	Alashahrani 22 [5]	TSK score in the range of 17–68
	Ohji 22 [29]	TSK-11 score
	Reinking 22 [20]	TSK-17 score in the range of 17–68
	Slagers 21 [30]	TSK score in the range of 17–68
	Theuniessen 19 [16]	TSK-17 score in the range of 17–68
	Watanabe 23 [23]	TSK-11 score
	Bagheri 21 [25]	TSK score
	Fukano 20 [21]	TSK-17 score in the range of 17–80
	Hart 19 [26]	TSK score in the range of 17–68
	Houston 14 [22]	TSK-11 score
	Jedvaj 21 [24]	TSK-17 score
	Kvist 04 [28]	TSK score
Coping strategies	Bagheri 21 [25]	Coping strategies questionnaire—27 items, categorized into 6 domains scored separately
Injury-related fear avoidance	Fukano 20 [21]	AFAQ score in the range of 10–50
	Houston 14 [22]	Fear Avoidance Beliefs Questionnaire
Knee confidence	Hart 19 [26]	(VAS) 0–10 and KOOS quality-of-life subscale
Psychological readiness to return to sport	Hart 19 [26]	ACL Return-to-Sport after Injury Scale, 0–100
Reinjury anxiety	Houston 14 [22]	Reinjury anxiety inventory, 28 items
Patient-reported fear	Paterno 18 [11]	TSK-11 score in the range of 11–44

TSK = Tampa Scale of Kinesiophobia; AFAQ = Athlete Fear Avoidance Questionnaire; VAS = Visual Analog Scale; KOOS = Knee Injury and Osteoarthritis Outcome Score; ACL = Anterior Cruciate Ligament.

## Data Availability

The data that support the findings of this study are available from the corresponding author, J.P.A., upon reasonable request.

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
