# Peer review of "Kinesiophobia in Injured Athletes: A Systematic Review"

_jfmk, 2024, doi:10.3390/jfmk9020078_

Round 1

Reviewer 1 Report

Comments and Suggestions for Authors

I really appreciate the opportunity to review this manuscript entitled “Kinesiophobia in Injured Athletes: A Systematic Review” This is important to assess kinesiophobia in this population.  I remark some issues (most of them in methods) in order to improve the quality of this manuscript.

Abstract is clear but should be improve, in the conclusions epigraph, ideas that do not emerge from the study for example “Clinicians should implement proper education, recognition, assessment, and treatment plan to help athletes with Kinesiophobia” should be removed (it was not a goal of the review).

Introduction was well structure and shows the necessity for this research. The aim of the paper is clear at the end of the introduction.  

At the methods section, there are some questions that should be review. About inclusion criteria, why do you choose all injured athletes? If in the introduction you are exposing that it is necessary and update, why do not choose the last 5-10 years? About quality assessment of the papers, why do you concrete what kind of study design are included and this is not clear in the inclusion criteria?

About results is the same question as methods. Table 4, it would be interesting to concrete what kind of injured athletes have. Discussion summarize and explain in a good way the finding. In future line you talked about the lack of studies with upper extremity injuries in athletes, you should explain that before, in your methods and results. Conclusions were correct but can be more concise as it has been said in the abstract.

Author Response

REVIEWER 1

I really appreciate the opportunity to review this manuscript entitled “Kinesiophobia in Injured Athletes: A Systematic Review” This is important to assess kinesiophobia in this population.  I remark some issues (most of them in methods) in order to improve the quality of this manuscript.

Response Thank you.

Abstract is clear but should be improve, in the conclusions epigraph, ideas that do not emerge from the study for example “Clinicians should implement proper education, recognition, assessment, and treatment plan to help athletes with Kinesiophobia” should be removed (it was not a goal of the review).

 Response Thank you for pointing this out. We have removed the above mentioned information . We have added information about study quality to stay consistent with what emerged from the study.

Introduction was well structure and shows the necessity for this research. The aim of the paper is clear at the end of the introduction. 

 Response Thank you.

At the methods section, there are some questions that should be review. About inclusion criteria, why do you choose all injured athletes?

Response: Thank you for the comment. As our purpose was to examine Kinesiophobia specifically in injured athletes, we chose that criteria for inclusion.

If in the introduction you are exposing that it is necessary and update, why do not choose the last 5-10 years?

Response: Thank you. We chose a longer timeframe to not miss important information. In the revised manuscript, we have added information in the Clinical Implications and Application section as follows: The current review provides evidence that there is an increasing amount of interest around the topic of Kinesiophobia in injured athletes, evidenced by the finding that in the final included articles, almost all  (13 of 14) of final included articles are within the last 10 years.

About quality assessment of the papers, why do you concrete what kind of study design are included and this is not clear in the inclusion criteria?

Response: Thank you. We chose to not exclude any manuscripts on the basis of study design in our initial filtering so that we could get more initial articles. We wanted to provide information about the current status of the research in the area of Kinesiophobia in injured athletes. Thus, we provide information about the type of study designs in the current manuscript. As per reviewer suggestions, we have added the following information in the revised manuscript as follows: The articles in this review included cross-sectional, prospective cohort, and a randomized controlled trial. This finding indicates that there is increasing interest in the area with researchers examining Kinesiophobia in injured athletes using multiple types of study designs.

About results is the same question as methods. Table 4, it would be interesting to concrete what kind of injured athletes have. Discussion summarize and explain in a good way the finding.

Response: Thank you for the comment. We have added kinds of injured athletes in Table 4 and added information in the revised text as follows: The majority of the researchers have examined Kinesiophobia in athletes with lower extremity injurie. Specifically, several authors have examined Kinesiophobia in athletes with knee injuries including Anterior Cruciate Ligament (ACL) injury being most common knee injury, supporting the idea that ACL injury and reconstuction is extensively associated with Kinesiophobia. Several authors also examined Kinesiophobia in athletes with ankle instability, which is understandable given that lateral ankle sprain is the most prevalent lower extremity musculoskeletal injury in physically active individuals.

In future line you talked about the lack of studies with upper extremity injuries in athletes, you should explain that before, in your methods and results. Conclusions were correct but can be more concise as it has been said in the abstract.

Response: Thank you. We have added information in the discussion limitations section as follows: For example, we did not find studies assessing Kinesiophobia for athletes with upper extremity injuries, with only one study examining musculoskeletal injuries irrespective of location.

Reviewer 2 Report

Comments and Suggestions for Authors

The topic of the article is extremely important and interesting, both for athletes after an injury, as well as for their partners (coaches, parents). I miss the comparison between different sports, but as explained in limitations, this would be a topic for further research.

I suggest perhaps one detail to be better explained:

Lines 260-267: TSK test is the most popular measurement tool to assess Kinesiophobia, but in lines 202-205 you are explaining that the TSK-J was not a recommended patient-reported outcome measure for psychological symptoms because of low validity and responsiveness.? Perhaps explain in Discussion more precisely. Thank you.

Limitations of the study are well written, those are excellent suggestions for future research.

Author Response

The topic of the article is extremely important and interesting, both for athletes after an injury, as well as for their partners (coaches, parents). I miss the comparison between different sports, but as explained in limitations, this would be a topic for further research.

 Response Thank you. Yes – we agree that this is an important topic for further research.

I suggest perhaps one detail to be better explained:

Lines 260-267: TSK test is the most popular measurement tool to assess Kinesiophobia, but in lines 202-205 you are explaining that the TSK-J was not a recommended patient-reported outcome measure for psychological symptoms because of low validity and responsiveness.? Perhaps explain in Discussion more precisely. Thank you.

 Response Thank you.  We have added discussion as follows: Although  Huang et al indicate that the Japanese version of the TSK (TSK-J) may not the best way to assess psychological factors for patients with ACL injury, most other researchers indicate the TSK as a means to objectively measure Kinesiophobia.

Limitations of the study are well written, those are excellent suggestions for future research.

 Response Thank you.